# Water Treatment in Hybrid Connection of Coagulation, Ozonation, UV Irradiation and Adsorption Processes

**Beata Karwowska \*, Elżbieta Sperczyńska and Lidia Dąbrowska**

Faculty of Infrastructure and Environment, Czestochowa University of Technology, Dąbrowskiego St. 69, 42-201 Częstochowa, Poland; elzbieta.sperczynska@pcz.pl (E.S.); lidia.dabrowska@pcz.pl (L.D.)
\* Correspondence: beata.karwowska@pcz.pl; Tel.: +48-34-3250-491; Fax: +48-34-3250-101

**Abstract:** In recent years, conventional water treatment systems have been supported by ozonation or UV irradiation processes. The efficiency of four hybrid processes: (1) coagulation and adsorption, (2) ozonation and coagulation, (3) ozonation, coagulation and adsorption, (4) ozonation, UV irradiation, coagulation and adsorption of inorganic and organic pollutants removal was analysed. In the presented study, the content of organic matter in natural water was evaluated as colour, oxidisability (OXI), total and dissolved organic carbon (TOC and DOC) content and UV absorbance at the wavelength of 254 nm for natural and modified water. Additionally, removal of $Ni^{2+}$, $Cd^{2+}$ and $Pb^{2+}$ ions during the treatment processes was analysed. The coagulation process with the use of polyaluminium chloride removed 45% of colour and 39, 26% and 45% of OXI, TOC and UV absorbance, respectively. Using the ozonation before coagulation increased efficiency of colour and $UV_{254}$ absorbance reduction by 33% and 25%, respectively. Coagulation with both UV irradiation and adsorption had insignificant results on the analysed factors value. The coagulation process was the most efficient for metal ions removal (40–78%). The ozonation process before coagulation increased removal up to 55–85%. Additional irradiation with UV or using of the adsorbent during coagulation of initially ozonated water had an insignificant impact on metal ions concentration in water.

**Keywords:** hybrid processes; water treatment; coagulation; adsorption; ozonation; UV irradiation; heavy metals; organic matter

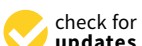

## 1. Introduction

One of the common water contaminations are heavy metal ions, which have a human health impact due to their real toxicity for organs as well as persistence and accumulation in the biological food chain [1]. The concentration of heavy metals in water is relatively very low when compared to the bottom sediments or sewage sludge. The evaluation of their content in aquatic media as sediment, benthic organisms and fishes is a major environmental target. Generally, the metals in minerals and rocks are not dangerous, but they are much more toxic when they are dissolved in water. Zhou et al. stated that concentrations of heavy metals in water are generally lower in developed regions (Europe and North America) than in developing ones (Africa, Asia, and South America) [2]. Over time, the main sources of heavy metal pollution in natural water have changed from primary mining and manufacturing to metal containing waste discharge. Regional determination of heavy metal ions content should concentrate on specific sources. Evaluation of metal emissions standards, limiting metal contents in industrial products, and pretreatment of metal should be proven avenues for controlling heavy metal pollution in rivers and lakes [2]. Surface and groundwater are the major source of drinking water and are also used for irrigation. Heavy metal ions, for example nickel, cadmium and lead are limited in water for human consumption. In the European Union Directive, their permissible concentrations are 0.02, 0.005 and 0.01 mg/L, respectively [3]. The type of water contamination and its potential application indicates processes used for water treatment.

The coagulation process is commonly used to remove inorganic and organic colloidal impurities as well as hard-to-settle suspensions from water. The effect of efficient coagulation is to reduce water turbidity and colour. In the coagulation process, the most commonly used reagents are aluminium and iron salts, both non-hydrolysed and pre-hydrolysed. Pre-hydrolysed coagulants contain hydroxyl groups which determine their increased basicity and include, among others, polyaluminium and polyferric (III) chlorides and sulphates. The use of, e.g., polyaluminium chlorides may be more efficient in destabilizing negative colloids, which mainly cause turbidity and colour of water, compared to the effects achieved with the use of aluminium sulphate due to the presence of polymeric forms of aluminium in the solutions of these coagulants. Also, their use may be useful in lowering natural organic matter (NOM) and heavy metal ions in water [4,5]. The degree of removal of organic substances increases with their content in the treated water, also with their molecular weight and degree of aromaticity. For this reason, macromolecular hydrophobic compounds absorbing UV rays, which are the main group of precursors of by-products of water chlorination, e.g., trihalomethanes (THM), are best removed. According to the authors of various publications [5–9], during the coagulation process, the reduction of turbidity and colour may be from 50 to 95%, and the efficiency of removal of organic compounds determined as total dissolved organic carbon (DOC) and UV absorbance in the range of 25–55% and 40–80%, respectively. The THM forming potential may be reduced by 25–65% [6].

In order to reduce the content of organic compounds and heavy metal ions in water, in addition to the coagulation process, other technological processes, such as adsorption, ion exchange, oxidation and membrane processes can be used. Adsorption is a commonly used process in water treatment. There are numerous literature reports on this topic. They include the use of various types of activated carbons, removing both natural organic matter [10] and the entire spectrum of metals, including heavy metals [11]. Activated carbon has become the most promising adsorbent for heavy metals from water because of its high potential absorbing capacity with respect to removal of different pollutants [1,12,13]; what is more, modification of commercial activated carbon improves adsorbent capacities for pollutants including heavy metals [14]. Efficiency of metals removal from water is influenced by several factors such as pH, temperature, concentration of metals, adsorbent dose and time of process [1,13].

The most commonly used activated carbon in the adsorption process is in the granular or powdered form [15]. Granular carbons constitute separate beds of filter filling or one of the layers of the bed. Powdered carbons are used periodically when the water quality deteriorates and are most often introduced during the coagulation process. This treatment method has many advantages because it can be implemented in various technological variants, it is easy to use, flexible and has a low energy requirement. In the adsorption process, not only natural organic substances and heavy metal ions, but also halogen derivatives of organic compounds, dyes, pharmaceuticals, and toxins can be efficiently removed [16].

The efficiency of removing poorly adsorbable compounds can be increased by applying ozonation before adsorption. Ozone can react with substances dissolved in water in two ways: in the direct reaction of molecular ozone and indirectly through the reactions of free radicals formed during decomposition of the ozone molecule in water. The latter method is carried out mainly by generating free hydroxyl radicals, the reactions of which are non-selective and the oxidative potential is very high. Although ozonation does not lead to the mineralization of organic carbon, but rather a slight decrease in its content is observed, under neutral pH conditions, the amount of high molecular weight fractions is significantly reduced and the amount of low molecular weight fractions increases. The most important effect of ozonation is the increase in the content of biodegradable organic matter. A significant reduction in the total organic carbon (TOC) content in water, as shown by Chin and Bérubé [17] and Petronijević et al. [18], can be achieved during a photochemical process, the simultaneous use of ozonation and UV radiation.

The advantages and disadvantages of coagulation, PAC (powdered activated carbon) adsorption, ozonation and UV irradiation used for water treatment in the study are presented in Table 1.

**Table 1.** Selected advantages and disadvantages of coagulation, PAC (powdered activated carbon) adsorption, ozonation and UV irradiation.

| Process | Advantages | Disadvantages |
| --- | --- | --- |
| Coagulation | Simple change of coagulant type and dose in dependency on water pollution type | Post-coagulation sludge formation Neccesary of additional sedimentation and filtration processes |
| PAC adsorption | Simple change of PAC type and dose in dependency on water pollution type Possibility of introduction in different location of the technology system | Problems with separation of PAC and water PAC not regenerable |
| Ozonation | Strong oxidative behaviour (high redox potential) | Possibilities of by-products formation High exploitative costs |
| UV irradiation | No by-products formation Simple exploitation | Limitation for process driving: water turbidity, colour and hardness |

The aim of the study was the evaluation of the usefulness of the selected adsorbent for heavy metal ions removal from water as well as determination of water treatment efficiency with respect to removal of turbidity, color, organic substances and heavy metal ions as a result of application of hybrid connection of the ozonation, UV irradiation, coagulation and adsorption processes.

## 2. Materials and Methods

### 2.1. Materials

Surface water from the Stradomka River collected in the city of Częstochowa (Poland) was used for the research. The research was carried out on both natural surface water and modified surface water. The modification consisted in introducing heavy metal ions into the water in the form of nitrate solutions, in an amount providing the concentration of individual metal ions of approximately 1.0 mg/L. The model water, which was distilled water enriched with the same heavy metal ion solutions as surface water, was also used in the research into the adsorption process. The initial concentrations of any metal ion individually in the model water were 0.5 and 1.0 mg/L.

The highly alkaline, pre-hydrolysed polyaluminium chloride with the trade name PAX-XL19F (PAX), produced by the Kemipol company in the city of Police (Poland), was used as a coagulant. The coagulant solution was characterized by a basicity of 85% ($OH^-/Al^{3+}$ = 2.55) and the content of $Al_2O_3$ of 16.1%. The coagulant and its dose were selected on the basis of the results of the previous research [19]. The used adsorbent was powdered activated carbon CWZ-30 produced by the GRYFSKAND company in the city of Hajnówka (Poland). Carbon had a specific surface area of 1134 $m^2$/g, the iodine value of 1190 mg/g and the methylene number of 30 mL. Ninety percent of the carbon grain size was below 0.06 mm.

### 2.2. Methods

In the first stage of the research, the process of water ozonation and irradiation with UV radiation was carried out in laboratory conditions. Ozone was produced by a laboratory generator (Model L20 SPALAB, Korona, Poland) from high-purity (99%) oxygen and 1.7 L of surface water were introduced into the reactors (cylinders with a volume of 2 L). Water ozonation was carried out at a constant ozone flow rate of 1.5 L/min, and the ozone dose was adjusted by changing the ozonation time, which was 4, 8 and 12 min. The ozone dose introduced into the water within 1 min was approximately 2 $mgO_3$/L. A spherical stone diffuser at the bottom of the reactors was used to disperse ozone.

For UV irradiation, a low-pressure mercury lamp with a power of 15 W, generating light with a wavelength of 253.4 nm, placed in a laboratory reactor with a volume of 0.8 L was used. Illumination was carried out continuously for 5, 10 and 20 min.

The adsorption process was carried out in the second stage. For this purpose, powdered activated carbon (PAC) was added in doses of 25, 50, 100 mg/L to 100 mL of model water. The samples were shaken for 15, 30 and 60 min with a frequency of 125 cycles/min. After shaking, the samples were filtered through a filter paper. The obtained filtrates were acidified with 0.2 mL concentrated nitric acid and the concentration of heavy metal ions was determined by the AAS (Atomic Absorption Spectrometry) method.

In the third stage, the modified surface water was purified in the coagulation process (1) and in hybrid systems combining unit processes: coagulation + adsorption (2); ozonation + coagulation (3); ozonation + coagulation + adsorption (4); ozonation + UV irradiation + coagulation + adsorption (5).

The doses of the used reagents were: PAX coagulant-3 mgAl/L, PAC-50 mg/L. Ozonation was carried out during 10 min with a dose of 2 mgO$_3$/L of water in a minute. Irradiation with a low-pressure UV lamp was carried out for 10 min.

The tests were carried out by measuring 1.7 L of modified water for six beakers: non-ozonized water for two beakers, ozonized water for another two beakers and ozonized water, irradiated with UV radiation for the last one.

The coagulant solution was introduced to the water in all the beakers and the beakers were placed in a stand with stirrers. The solutions were mixed for 1 min using 300 rpm. Powdered activated charcoal was then added to the water in beakers 2, 4 and 5 and the water in all beakers was further stirred for 2 min. After rapid mixing was completed, the rotational speed was reduced to 30 rpm and mixing continued for 15 min. The samples were left in the beakers for a 60 min sedimentation period.

Then 0.3 L of water was decanted. The following determinations were made in both raw and decanted water: pH, turbidity, colour, oxidisability, TOC, DOC, UV absorbance, heavy metal ions (Ni$^{2+}$, Cd$^{2+}$, Pb$^{2+}$) concentration.

The schematic diagrams of all technologies used for water treatment are presented in Figure 1.

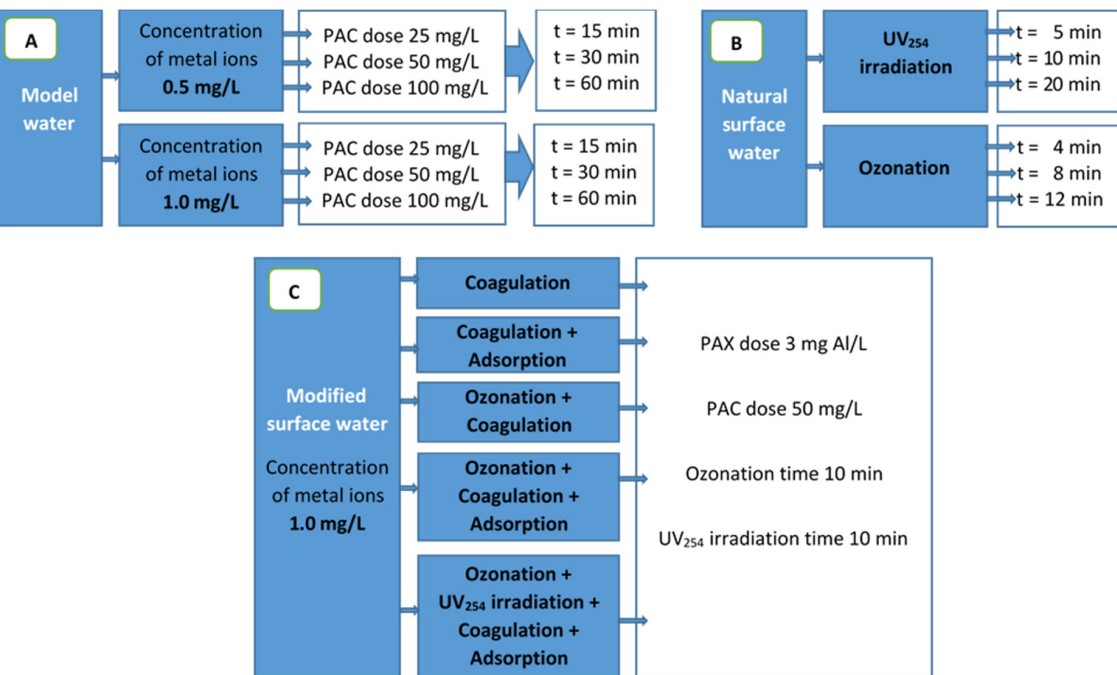

**Figure 1.** Schemes of treatment method applied to different types of analysed water: (**A**)—model water, (**B**)—natural surface water, (**C**)—modified surface water; PAX—pre-hydrolysed polyaluminium chloride, PAC—powdered activated carbon.

*2.3. Analytical Procedure*

The surface water quality indicators, before and after the coagulation process, were determined using the following methods: pH—potentiometric (pHmeter HI2002, Hanna Instruments, Poland), turbidity-nephelometric (turbidity meter HI 98703, Hanna Instruments, Poland), colour by comparing with standards on the platinum-cobalt scale, oxidisability-permanganate, TOC and DOC (after filtering the water through a membrane filter 0.45 μm), by high-temperature oxidation in a stream of oxygen and $CO_2$ measurement with an NIDR detector (carbon analyser Vario TOC Cube, Elementar, Germany), UV absorbance at the wavelength of 254 nm (cuvette 1 cm) using a spectrophotometer UV 5600 (Shanghai Metash Instruments, China), heavy metal ions (nickel, lead and cadmium) by atomic absorption spectrometry (spectrometer novAA 400, Analytik Jena, Germany).

*2.4. European Union Legislative Limits of Selected Indicators of Water Quality*

Table 2 presents the water quality standards adopted in the European Union for selected indicators [3].

**Table 2.** Studied parameters values in the European Union Council Directive.

| Parameter | Parameter Value, Unit | Notes |
| --- | --- | --- |
| Hydrogen ion concentration | $pH \geq 6.5$ and $pH \leq 9.5$, pH units | The water should not be aggresive |
| Colour | Acceptable to consumers and no abnormal change | |
| Turbidity | Acceptable to consumers and no abnormal change | Member States should strive for a parametric value not exceeding 1.0 NTU |
| Oxidisability | 5 $mgO_2$/L | |
| TOC | No abnormal change | |
| Cadmium | 0.005 mg/L | |
| Lead | 0.010 mg/L | |
| Nickel | 0.020 mg/L | |

TOC (total organic carbon).

## 3. Results and Discussion

The surface water was characterized by a colour ranging from 60 to 90 mgPt/L, and $UV_{254}$ absorbance of 0.285–0.519 1/cm. The content of organic compounds determined as oxidisability (OXI), TOC and DOC varied respectively in the ranges: 6.9–13.0 $mgO_2$/L; 6.9–11.8 mgC/L; 6.3–11.2 mgC/L.

The results of the effect of ozonation time or UV irradiation of surface water on the colour change, TOC and DOC content as well as the values of OXI and $UV_{254}$ absorbance are presented in Tables 3 and 4, and the percentage reduction in the values of these indicators is shown in Figure 2.

**Table 3.** Values of selected parameters of raw natural water and water after ozonation process, ozone dose 2 $mgO_3$/L.

| Parameter | Unit | Raw Water | Ozonation Time, Minutes | | |
| --- | --- | --- | --- | --- | --- |
| | | | 5 | 8 | 12 |
| Colour | mgPt/L | 90 | 74 | 62 | 57 |
| OXI | $mgO_2$/L | 12.5 | 10.6 | 9.9 | 9.5 |
| TOC | mgC/L | 11.8 | 11.6 | 11.4 | 11.0 |
| DOC | mgC/L | 11.2 | 10.7 | 10.7 | 10.6 |
| $UV_{254}$ | 1/cm | 0.497 | 0.389 | 0.326 | 0.258 |

OXI (oxudisability), DOC (dissolved organic carbon)

**Table 4.** Values of selected parameters of raw natural water and water after $UV_{254}$ irradiation (low-pressure lamp).

| Parameter | Unit | Raw Water | Irradiation Time, Minutes | | |
|:---:|:---:|:---:|:---:|:---:|:---:|
| | | | **5** | **10** | **20** |
| Colour | mgPt/L | 90 | 90 | 85 | 80 |
| OXI | $mgO_2/L$ | 13.0 | 12.3 | 11.9 | 11.5 |
| TOC | mgC/L | 11.5 | 11.7 | 11.2 | 10.9 |
| DOC | mgC/L | 11.1 | 11.0 | 10.8 | 10.4 |
| $UV_{254}$ | 1/cm | 0.519 | 0.472 | 0.431 | 0.382 |

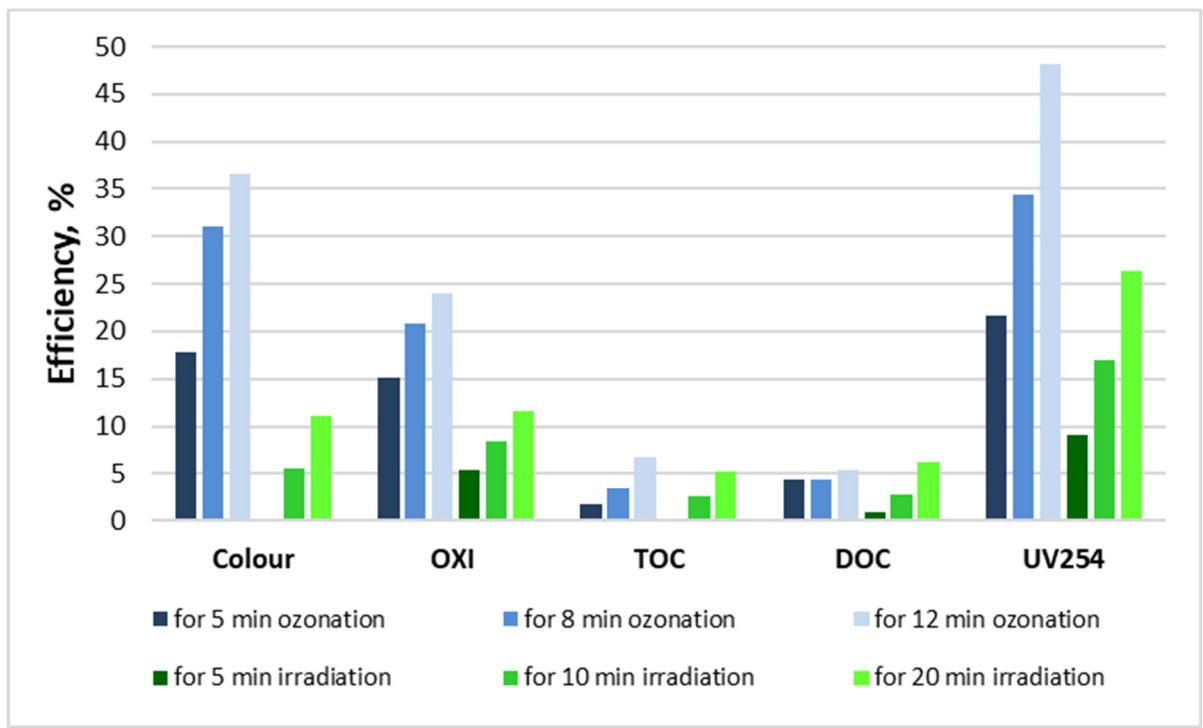

**Figure 2.** Efficiency of selected parameters of raw natural water removal after ozonation and UV irradiation processes.

Ozonation was found to reduce the colour and $UV_{254}$ absorbance value. The reduction was, depending on the ozonation time, from 18 to 37% and from 22 to 48%, respectively. The value of the OXI index decreased by 15–24%. No significant changes in the content of TOC and DOC (maximum 5–7%) were found. This proves that the total content of organic matter in the water changed only slightly after ozonation. On the other hand, the quality of organic compounds changed, as a result of the partial decomposition of some organic substances—the content of carbon in aromatic chemicals was reduced. It was indicated by a significant decrease in the value of $UV_{254}$ absorbance after water ozonation. As Zainudin et al. [20] reported, ozonation generally increases the biodegradability of NOM in water by transforming larger organic molecules into smaller ones that are more easily biodegradable. Similar results were obtained by Sadrnourmohamadi et al. [21]. The authors showed that the use of ozone in a dose of 1 $mgO_3/1$ mgC made it possible to reduce the DOC content and $UV_{254}$ absorbance values by 42% and 95%, respectively.

After the process of irradiation with UV radiation, no significant changes in the decrease in the water colour and the overall content of organic compounds were found. The greatest reduction in both the colour and oxidisability occurred after the irradiation time of 20 min and amounted to 11% and 12%. For the longest irradiation time, organic compounds designated as TOC and DOC decreased only by 5% and 6%. On the other

hand, the value of $UV_{254}$ absorbance decreased by 17% with the irradiation time of 10 min, with the time extended to 20 min, by 26%.

The effect of the use of adsorption processes on the analysed heavy metal ions concentration in model water is presented in Tables 5 and 6 as well as in Figure 3 and in modified water in Table 7.

**Table 5.** Metal ions concentration (mg/L) in model water after adsorption process with PAC, initial metal ions concentration: 0.5 mg/L.

| PAC Dose mg/L | $Ni^{2+}$ Ions Concentration Contact Time, Minutes | | | $Cd^{2+}$ Ions Concentration Contact Time, Minutes | | | $Pb^{2+}$ Ions Concentration Contact Time, Minutes | | |
|---|---|---|---|---|---|---|---|---|---|
| | 15 | 30 | 60 | 15 | 30 | 60 | 15 | 30 | 60 |
| 25 | 0.43 | 0.41 | 0.38 | 0.35 | 0.34 | 0.34 | 0.27 | 0.26 | 0.25 |
| 50 | 0.41 | 0.37 | 0.36 | 0.35 | 0.34 | 0.33 | 0.27 | 0.25 | 0.23 |
| 100 | 0.35 | 0.34 | 0.34 | 0.34 | 0.34 | 0.31 | 0.27 | 0.23 | 0.18 |

**Table 6.** Metal ions concentration (mg/L) in model water after adsorption process with PAC, initial metal ions concentration: 1.0 mg/L.

| PAC Dose mg/L | $Ni^{2+}$ Ions Concentration Contact Time, Minutes | | | $Cd^{2+}$ Ions Concentration Contact Time, Minutes | | | $Pb^{2+}$ Ions Concentration Contact Time, Minutes | | |
|---|---|---|---|---|---|---|---|---|---|
| | 15 | 30 | 60 | 15 | 30 | 60 | 15 | 30 | 60 |
| 25 | 0.91 | 0.91 | 0.90 | 0.92 | 0.90 | 0.86 | 0.77 | 0.76 | 0.74 |
| 50 | 0.91 | 0.91 | 0.88 | 0.91 | 0.90 | 0.82 | 0.66 | 0.62 | 0.59 |
| 100 | 0.90 | 0.87 | 0.86 | 0.90 | 0.88 | 0.79 | 0.60 | 0.57 | 0.50 |

The nickel ion removal rate from the model water, at an initial concentration of 1.0 mg/L, was 10–14% for the 100 mg/L PAC dose, 9–12% for the 50 mg/L dose and 9–10% for the 25 mg/L dose. At the initial nickel concentration of 0.5 mg/L, removal of 30–32%, 20–28% and 18–20% was obtained, respectively, for the 100, 50 and 25 mg/L PAC doses. The results indicate that the changes in the metal removal efficiency were not large when PAC dose increased, and the nickel itself was adsorbed to the tested material to a small extent. As the duration of the process was extended, the efficiency of the process tended to increase, but the changes were small, without a significant effect on the overall efficiency of the powdered activated carbon adsorption process. Removal of cadmium from water followed similar processes as in the case of nickel. Cadmium was slightly adsorbed. The removal rate was 10–20% at the initial concentration of 1 mg/L, and about 38% at the initial concentration of 0.5 mg/L. The best results were obtained with the use of PAC in a dose of 100 mg/L, for the initial metal concentration of 0.5 mg/L and the duration of the process 60 min. The process efficiency was then 38%. The effect of extending the process time on the increase in the efficiency was the most apparent in the case of the solution with the initial concentration of metal of 1 mg with time change from 30 to 60 min. The process efficiency increased then from 12 to 21%.

Among the examined heavy metals, lead was the most susceptible to adsorption. The observed efficiency of the adsorption process under the conditions of the conducted experiment reached the level up to approximately 60%. Better removal results were found for a lower initial concentration of metal ions in the analysed water. The extension of the process time resulted in a slight increase in the lead concentration lowering effect, like in the case of nickel and cadmium. For the initial concentration of 1 mg/L, the removal efficiency varied from 40 to 43 and 50% for the process times: 15, 30 and 60 min, respectively, after using PAC in a dose of 100 mg/L. For the initial concentration of 0.5 mg/L, the percentage removal efficiency was slightly higher and amounted to 46, 54 and 64%, respectively for process times: 15, 30 and 60 min in a dose of 100 mg/L. The promising results for the removal of $Pb^{2+}$ ions were also reported by Lavecchia et al. during adsorption on the

novel natural materials [22]. Rapid adsorption of heavy metals on activated carbon was demonstrated previously [13]. Authors studied change in removal of metals on activated carbon in dependency on time. They detected about 40% efficiency in the first 30 and 60 min of the process and after 180 min it did not increase significantly. Additionally, increasing of the adsorbent dose was a factor enhancing process efficiency. The effect of the increased adsorbent dosage as well as time of process was more significant in the case of lead removal than other analysed metals: nickel and cadmium. Sun et al. [1] observed similar dependence. The increased adsorbent dosage significantly improved removal of the analysed heavy metals. They reported that optimal activated carbon dose was at the level of 0.8 g/L.

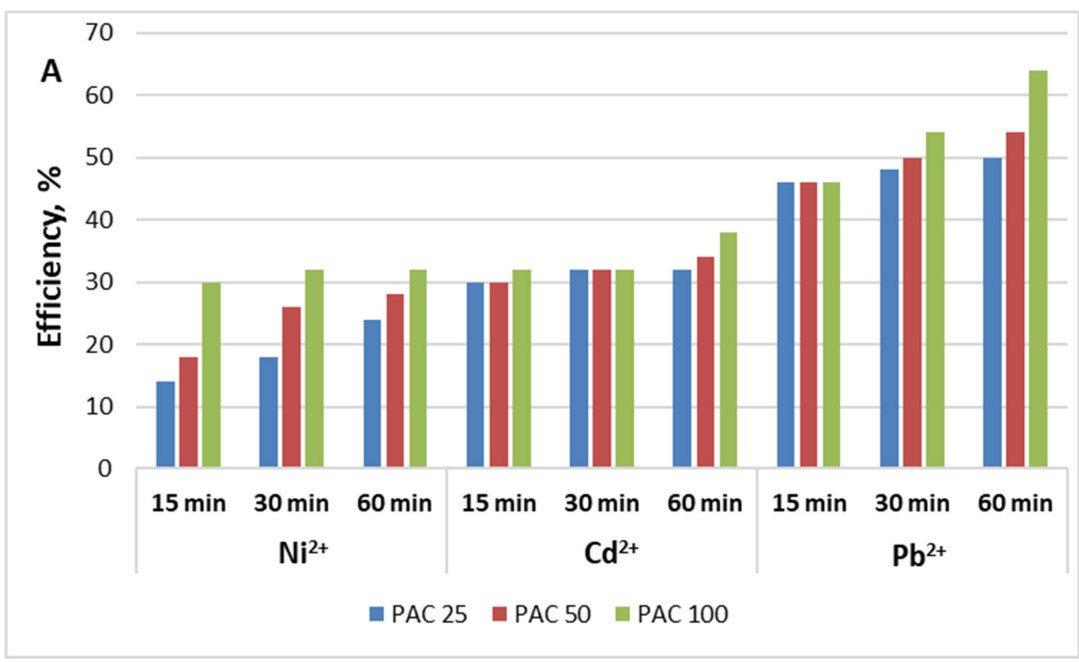

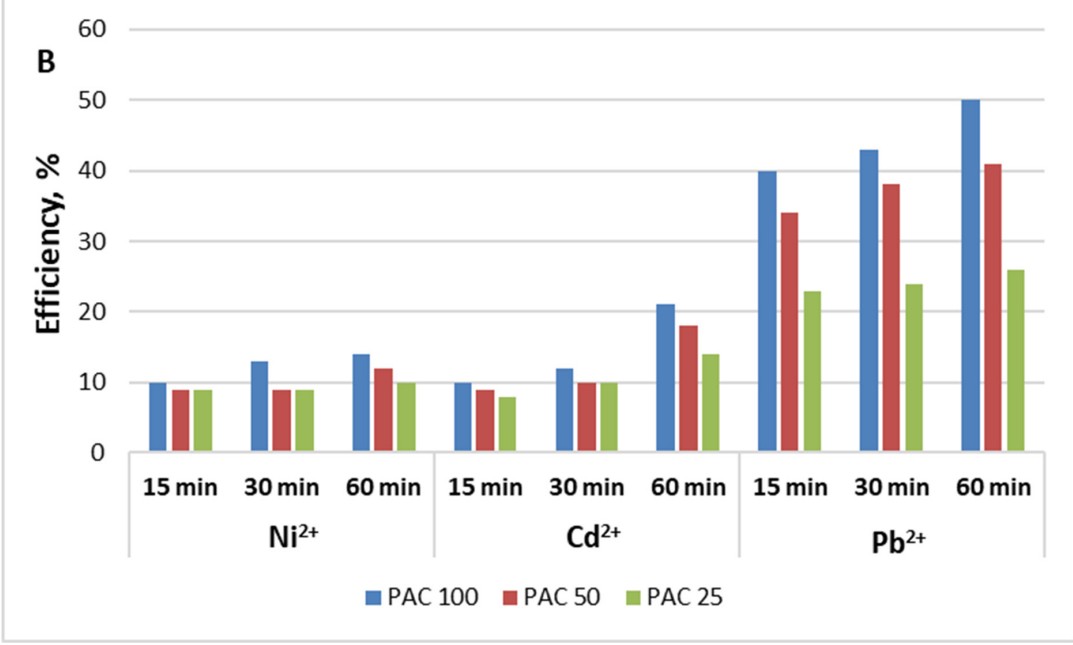

**Figure 3.** Efficiency of analysed metal ions removal from model water in the adsorption process with using powdered activated carbon (dose: 25, 50 and 100 mg/L), time of process 15, 30 and 60 min, initial concentration of heavy metal ions: (**A**) 0.5 mg/L and (**B**) 1.0 mg/L.

The next step of our study was analysis of metal adsorption on PAC from modified water. The results of the experiment are presented in Table 7. Nickel and cadmium were adsorbed from the modified surface water to powdered activated carbon with an efficiency of 24% and 33%, respectively. The obtained efficiencies were higher than in the case of model water (14% and 21%). The increase in the removal rate was approximately 10%. Lead, for which the removal degree was 45%, was the most susceptible to adsorption from the modified surface water. It was at a similar level as the one observed in the case of model water (50%).

**Table 7.** Metal ions concentration in modified water after adsorption on PAC (dose 100 mg/L), time of shaking 60 min, initial concentration of heavy metal ions 1.0 mg/L.

| Adsorbent | Metal Ions Concentration, mg/L | | |
|:---:|:---:|:---:|:---:|
| | $Ni^{2+}$ | $Cd^{2+}$ | $Pb^{2+}$ |
| PAC | 0.67 | 0.76 | 0.43 |

Adsorption techniques are used both to remove heavy metals from water and to recover valuable metals such as gold and silver. The adsorption capacity of activated carbons is very different, usually in the range from 30 to even 99%. Therefore, the results obtained during the research presented in the article, both in the case of model water and modified surface water, fit into this range of process efficiency reported in the literature. Kharrazi et al. [13] reported even 90% of activated carbon ability in removal of lead, Zaini et al. [14] 75% of lead and 50% of copper. Research studies are very often focused on the optimization of conducted adsorption process. The adsorption efficiency is largely dependent on the type of activated carbon used, the environment pH and the metal that is retained on the surface of the activated carbon. Most metals are better adsorbed when acidity increases. However, some, e.g., Cd (II), tend to increase adsorption with increasing pH to low-alkali ranges [11]. Sun et al. [1] studied the effect of pH on adsorption capacity towards selected heavy metals ions. Their results show enhancement of adsorption with increase of pH value, which indicated that high pH value is a beneficial parameter for heavy metal ions removal on activated carbon. Such evident behavior seems to be a promising target for future research in the area of heavy metals adsorption.

Changes in the values of surface water indicators (turbidity, colour, OXI, TOC, DOC, $UV_{254}$ absorbance) after the use of coagulation and coagulation in combination with other processes are presented in Table 8, and the percentage reduction in the values of these indicators in Figure 4.

**Table 8.** Values of selected parameters of raw natural water and treated water in the coagulation process and hybrid processes.

| Parameter | Unit | Raw Water | Treated Water with Processes | | | | |
|:---:|:---:|:---:|:---:|:---:|:---:|:---:|:---:|
| | | | C | C + A | $O_3$ + C | $O_3$ + C + A | $(O_3 + UV)$ + C + A |
| pH | - | 7.72 | 7.64 | 7.69 | 7.75 | 7.80 | 7.79 |
| Turbidity | NTU | 8.12 | 2.80 | 3.38 | 1.53 | 1.87 | 1.21 |
| Colour | mgPt/L | 60 | 20 | 15 | 10 | 5 | 5 |
| OXI | $mgO_2$/L | 7.6 | 4.5 | 4.4 | 4.0 | 3.9 | 3.5 |
| TOC | mgC/L | 6.9 | 5.1 | 4.9 | 4.7 | 4.1 | 3.5 |
| DOC | mgC/L | 6.3 | 4.6 | 4.2 | 4.3 | 3.5 | 3.1 |
| $UV_{254}$ | 1/cm | 0.85 | 0.149 | 0.099 | 0.085 | 0.053 | 0.044 |

C—coagulation; C + A—coagulation + adsorption; $O_3$ + C—ozonation + coagulation; $O_3$ + C + A—ozonation + coagulation + adsorption; $(O_3 + UV)$ + C + A—ozonation + $UV_{254}$ irradiation + coagulation + adsorption.

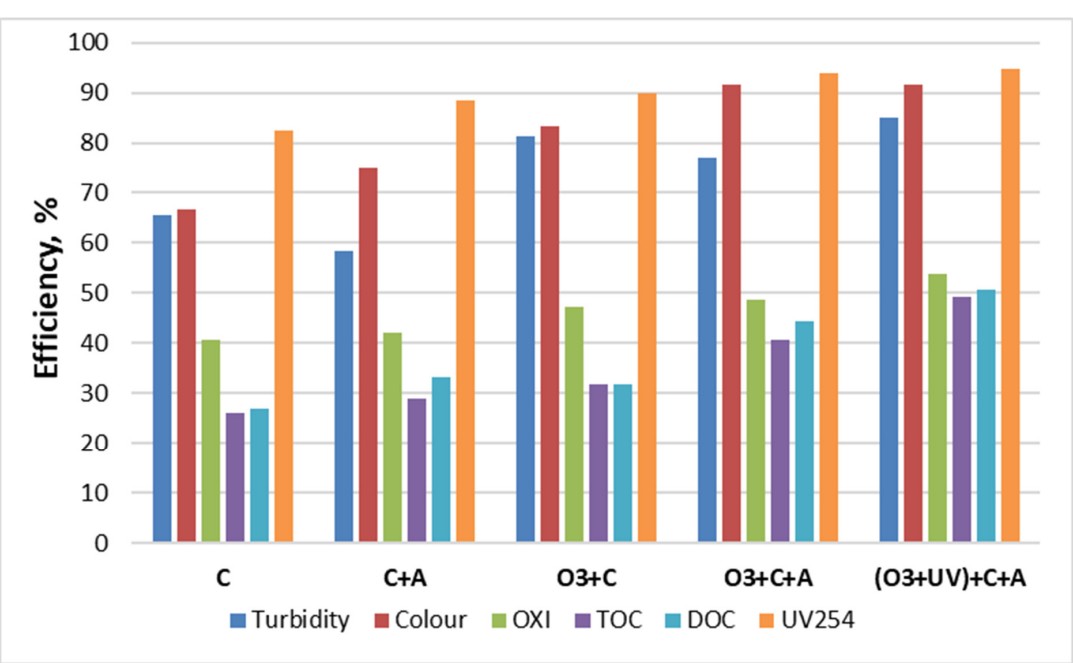

**Figure 4.** Efficiency of selected parameters removal from modified surface water treated in the coagulation process and hybrid processes; C—coagulation; C + A—coagulation + adsorption; $O_3$ + C—ozonation + coagulation; $O_3$ + C + A—ozonation + coagulation + adsorption; ($O_3$ + UV) + C + A—ozonation + $UV_{254}$ irradiation + coagulation + adsorption.

As a result of the coagulation process, the turbidity and colour were reduced by 66 and 67%, respectively. Supporting coagulation with the adsorption process with the use of PAC resulted in a colour reduction by the additional 8%, and supporting coagulation with ozonation by 16%. The use of combined ozonation + coagulation + adsorption processes allowed to reduce colour by 92% and turbidity by 87%. An acceptable level of water colour was achieved.

Removal of organic compounds determined by OXI was at a similar level in the case of coagulation and coagulation combined with adsorption (41–42%). During the coagulation preceded by ozonation, a 47% decrease in the value of this index was achieved, while the ozonation with UV irradiation + coagulation with a PAC adsorption system was applied by another 3%. Determined values (3.5–4.5 $mgO_2/L$) were lower than permissible (5 $mgO_2/L$) for water intended for human consumption. The degree of reduction in the content of organic compounds during the coagulations designated as TOC and DOC amounted to 26% and 27%, respectively. Supporting the coagulation process with adsorption or ozonation resulted in a further reduction of TOC and DOC by another 2–5% and 5–6%, respectively. Better effects were found when using extended hybrid systems: ozonation, coagulation + adsorption and ozonation with UV irradiation + coagulation + adsorption, during which the reduction of TOC and DOC content was 41–43% and 44–46%, respectively. According to Szlachta and Adamski [23], supporting coagulation by adsorption using PAC conducted at pH of approx. 6.0 can increase the efficiency of DOC removal from a few to 40%, and UV values from a few to 35%. Uyak et al. [24] obtained a 76% reduction in DOC content using a combined coagulation process using iron (II) sulphate and PAC. On the other hand, Kristiana et al. [25] showed in their studies that the use of combined coagulation processes with PAC adsorption resulted in the removal of NOM by 70%, which reduced the potential for THM formation by 80–95%.

Ozonation in combination with UV irradiation is an efficient catalytic process for the degradation of impurities that are difficult to decompose. The synergistic action of $O_3$ with UV radiation promotes the decomposition of ozone by direct and indirect production of hydroxyl radicals in water, which attack the aromatic ring in the molecule, resulting in the formation of smaller aliphatic molecules [26]. According to the literature data, the use

of initial ozonation increases the efficiency of removing organic compounds designated as DOC and UV [8,27]. The scope of changes depends on many factors, such as the composition of the treated water, ozone dose in relation to the content of DOC or the used coagulants. In tests conducted for groundwater, with the values of DOC and $UV_{254}$ absorbance, respectively 5.1 mgC/L and 0.211 1/cm, Tubić et al. [8] demonstrated that the initial ozonation of water allowed to increase the efficiency of removal of organic compounds by a maximum of 11% for DOC and 20% for $UV_{254}$ absorbance. Similar studies were carried out for surface water in Canada [27] using an iron (II) sulphate-based coagulant. The river water was characterized by high levels of alkalinity, hardness and DOC content. The studies showed that the efficiency of DOC removal during the coagulation at pH 8 for different doses of ozone and coagulant ranged from 32% to less than 40%.

The research on the assessment of the effect of ozonation and UV radiation, as unit processes and carried out in a combined manner, used for the initial preparation of water before the coagulation process was carried out by Li et al. [28]. The research results obtained by the authors confirmed the efficiency of the $O_3$, UV, $O_3$ + UV processes to reduce the content of organic compounds determined by the TOC index, whereby the $O_3$ + UV method was more efficient compared to using single processes. With the ozone dose of 1 $mgO_3$/1mgC and with the UV irradiation time of 60 min, the TOC content in water decreased by 38% and turbidity was lower than 1.0 NTU. The authors report that the initial water preparation had a positive effect on the removal of organic matter during the coagulation process. The application of UV with ozonation is believed to be more efficient in decomposition of THM precursors in waters with respect to the ozonation alone [20].

The effect of the use of single and combined water treatment processes on the reduction of the tested heavy metal ions concentration is presented in Table 9 and Figure 5.

Lead was best removed in the analysed processes. The removal efficiency ranged from 78% during the coagulation process to 86% for the combined processes of coagulation with ozonation, UV irradiation and adsorption with PAC. The changes in the lead removal efficiency were not very significant, but ozonation had the largest share in the intensification of the coagulation process. $Pb^{2+}$ ion concentration after ozonation and coagulation were at the same level as after the addition of activated carbon adsorption to these processes. Also, the additional irradiation of water with UV radiation did not increase the lead removal efficiency. Similar courses of concentration changes in the analysed waters were observed for the remaining metals, but the efficiency of the processes was lower. The efficiency of reducing the cadmium ions' concentration after the coagulation process was 51%, and after including adsorption with PAC-55%. A significant effect on the removal of these metal ions was found in the case of a combination of coagulation and ozonation. The obtained decrease in concentration was equal to 71%. The inclusion of additional processes—adsorption on activated carbon and UV irradiation—practically did not increase the efficiency of cadmium ion removal, which was 73% after the ozonation + coagulation + adsorption process and 72% after the additional UV radiation of water during ozonation.

**Table 9.** Heavy metal ions concentration (mg/L) in modified water treated with coagulation process and hybrid processes with using powdered activated carbon (dose 50 mg/L), initial heavy metal ions concentration 1.0 mg/L.

| Water Treated with Processes | Heavy Metal Ions Concentration, mg/L | | |
|:---:|:---:|:---:|:---:|
| | $Ni^{2+}$ | $Cd^{2+}$ | $Pb^{2+}$ |
| C | 0.60 | 0.49 | 0.22 |
| C + A | 0.56 | 0.45 | 0.18 |
| $O_3$ + C | 0.45 | 0.29 | 0.15 |
| $O_3$ + C+A | 0.45 | 0.27 | 0.15 |
| ($O_3$ + UV) + C + A | 0.43 | 0.28 | 0.14 |

C—coagulation; C + A—coagulation + adsorption; $O_3$ + C—ozonation + coagulation; $O_3$ + C + A—ozonation + coagulation + adsorption; ($O_3$ + UV) + C + A—ozonation + $UV_{254}$ irradiation + coagulation + adsorption.

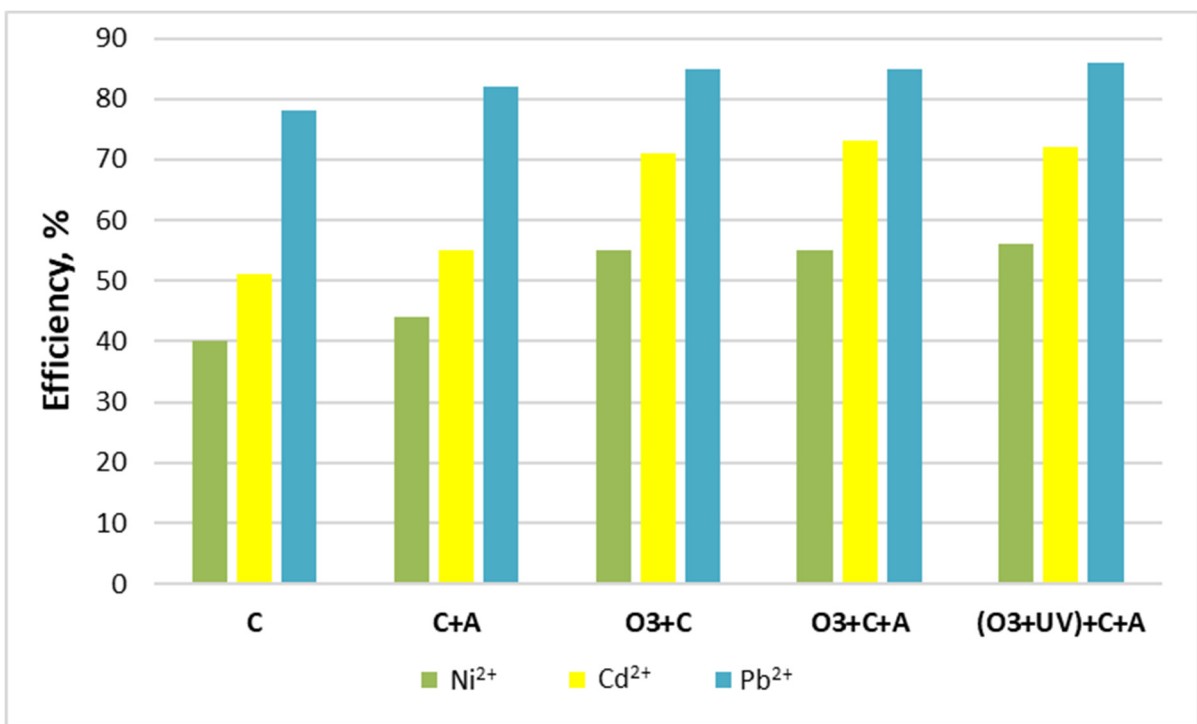

**Figure 5.** Efficiency of tested metal ions removal from modified surface water treated in the coagulation process and hybrid processes; C—coagulation; C + A—coagulation + adsorption; $O_3$ + C—ozonation + coagulation; $O_3$ + C + A—ozonation + coagulation + adsorption; $(O_3 + UV)$ + C + A—ozonation + $UV_{254}$ irradiation + coagulation + adsorption; PAC dose (50 mg/L), time of process 60 min, initial concentration of heavy metal ions: 1.0 mg/L.

Nickel susceptibility to removal during coagulation was lower than that of the previously discussed metals. The nickel concentration in water was reduced by 40% in the coagulation process. The addition of PAC did not have a significant effect on the removal of nickel ions, while the introduction of ozonation increased the efficiency to 55%. During the combined processes of ozonation + UV irradiation + coagulation + adsorption on PAC, a 57% reduction in nickel concentration was achieved. In the case of nickel ion removal, the slightly more significant effect of irradiation on their removal should be emphasized.

## 4. Conclusions

The usefulness of the coagulation process with the use of pre-hydrolysed salt for removing natural organic matter and heavy metal ions from surface water and supporting coagulation with the processes of ozonation, UV irradiation and adsorption on powdered activated carbon was demonstrated.

The efficiency of water treatment in the coagulation process with the use of polyaluminium chloride in the case of colour removal was 45%, organic compounds determined by the indicators of oxidisability, TOC and UV absorbance, respectively 39, 26% and 45%. A further increase in efficiency by 33% and 25% was found in the case of a decrease in colour and $UV_{254}$ absorbance, respectively, when using the ozonation process before coagulation. However, no significant changes in the total content of organic matter in water were found. Ozonation, therefore, did not lead to mineralization of organic carbon, but caused a change in the forms of organic compounds occurrence. Supporting coagulation with both UV irradiation and adsorption on PAC did not result in a significant increase in the reduction of the values of the analysed indicators.

The analysis of combined processes showed that, in the case of pollutants such as heavy metal ions, the coagulation process had the greatest share in their removal from water. Already after this stage of water purification, a 40–78% reduction in the concentration of heavy metal ions (nickel, cadmium and lead) was achieved. The intensification of this

process was observed after including the ozonation process before coagulation (55–85% removal), while the effect of additional irradiation with UV radiation was practically unnoticeable. Introduction of the adsorbent during coagulation of previously ozonized water basically had no effect on reducing the amount of heavy metal ions.

**Author Contributions:** Conceptualization, B.K., E.S. and L.D.; methodology, B.K., E.S. and L.D.; investigation, B.K., E.S. and L.D.; writing—original draft preparation, B.K., E.S. and L.D.; writing—review and editing, B.K., E.S. and L.D.; visualization, E.S.; supervision, B.K.; funding acquisition, B.K., E.S. and L.D. All authors have read and agreed to the published version of the manuscript.

**Funding:** The scientific research was funded by the statute subvention of the Czestochowa University of Technology, Faculty of Infrastructure and Environment.

**Institutional Review Board Statement:** Not applicable.

**Informed Consent Statement:** Not applicable.

**Data Availability Statement:** Not applicable.

**Conflicts of Interest:** The authors declare no conflict of interest.

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
