# Peer review of "Water Treatment in Hybrid Connection of Coagulation, Ozonation, UV Irradiation and Adsorption Processes"

_water, doi:10.3390/w13131748_

Round 1

Reviewer 1 Report

In general, it is a good research on a relevant topic. The text is quite easy to read and well structured. Nevertheless, it still needs some further improvements:

  • Line 30: "other biota" requires correction' as water and bottom sediments are not biota;
  • Lines 33-35: "Concentrations of heavy metals in water  are generally lower in developed regions (Europe and North America) than in developing  ones (Africa, Asia, and South America)": some Refr to this statement to prove it;
  • Line 37: What is "secondary metal waste discharge"?
  • Line 43: "permissible concentrations are 0.02; 0.005 and 0.01 mg/L respectively". Suggest to specify directly in the text that this is for EU;
  • Line 81: do you really mean "during the coagulation process"?
  • Line 110: "approx. 1.0 mg/L" - As a sum, or individually? for Ni, Cd, and Pb?
  • Line 139: "the ASA method" - provide a full name for this, and only then the abbreviation;
  • A suggestion for Methodology: add a scheme indicating treatment methods applied to different types (model/ modified) of water. This would help reader to quicker understand what types of water have been included in the experiments, and which treatment methods applied when to what;
  • Figure 1: The figure displays efficiency, not ozonation and irradiation time. Please correct the legend.
  • Line 193: Please express this in a more clear way: there were no significant changes for both TOC and DOC, so why do you state that organic carbon decreased?
  • Line 218: "that the changes in the metal removal efficiency were not large" - The changes not large: please specify, comparing what to what? At different PAC doses? Different contact time?
  • Figure 2: Please write "2+" as a superscript;
  • Figure 2: Suggest to indicate in figures themselves, which one is for 0.5 and which one for 1 mg/l. Or, mark the figures with a) and b)
  • Line 240-241: "The extension of the process time resulted in a slight increase in the lead ..." - Isn't it so also for Ni and Cd?
  • Line 237-256: Any ideas to explain why efficiency was the highest and the most increasing namely for Pb?
    Why efficiency was better for lower initial metal concentrations?
  • Lines 241 and 246: "The extension of the process time resulted in a slight increase ..." / "the extension of the experiment time resulted in an increase in ..." - You are repeating what has already been said;
  • Line 253: "than other previously analyzed metals" - what are those other previously analyzed metals?
  • Table 5: Suggest to place the table after the next paragraph, and make a more clear transition from describing model water and modified water.
    Why only this PAC dose and timing are presented for modified water?
  • Lines 265 - 280: What is particular about the type of activated carbon, or about the other factors of your research? So that other researchers, or practitioners could refer to your results in their work?
  • Line 293: "58" - It doesn't look like it is 58% in the table and the figure. You have either switched places C and C+A results there, or here in the explanation.
  • Editing of English language is required. It is actually quite uneven throughout the text: from a fluent text to some obvious mistakes.

Author Response

We would like to thank very much for the professional, helpful an useful Reviewers comments and suggestions. We are grateful for the Reviewer’s careful reading. Correction was made according to the comments. All changes in the manuscript are indicated by yellow color.

Answer for comments of Reviewer 1:

  • Line 30: "other biota" requires correction' as water and bottom sediments are not biota;

The sentence was corrected

  • Lines 33-35: "Concentrations of heavy metals in water  are generally lower in developed regions (Europe and North America) than in developing  ones (Africa, Asia, and South America)": some Refr to this statement to prove it;

The reference bibliography was attached and sentence was rearranged

  • Line 37: What is "secondary metal waste discharge"?

The sentence was rearranged to be simple and understandable

  • Line 43: "permissible concentrations are 0.02; 0.005 and 0.01 mg/L respectively". Suggest to specify directly in the text that this is for EU;

The sentence was corrected

  • Line 81: do you really mean "during the coagulation process"?

We would like to explain that the location for the introduction of the powdered activated carbon (PAC) is usually a raw water pipeline or a quick stirring tank, even before the application of the coagulant. Such location results in long contact time between PAC and water. PAC can also be applied with a delay in relation to the coagulant during coagulation process, which prevents carbon build-up in the formed agglomerates.

  • Line 110: "approx. 1.0 mg/L" - As a sum, or individually? for Ni, Cd, and Pb?

The given concentration was 1mg/L for any individual metal ion separately. The correction was made in the manuscript.

  • Line 139: "the ASA method" - provide a full name for this, and only then the abbreviation;

The full name of method and the proper English abbreviation were given.

  • A suggestion for Methodology: add a scheme indicating treatment methods applied to different types (model/ modified) of water. This would help reader to quicker understand what types of water have been included in the experiments, and which treatment methods applied when to what;

Thank you very much for the valuable suggestion. The relative schemes were introduced to the manuscript.

  • Figure 1: The figure displays efficiency, not ozonation and irradiation time. Please correct the legend.

The figure and legend were corrected (actual Figure 2).

  • Line 193: Please express this in a more clear way: there were no significant changes for both TOC and DOC, so why do you state that organic carbon decreased?

Thank you very much for the suggestion. The sentences are changed and more precisely describe the changes in organic matter transformation.

We would like to explain, that we wanted to write that total organic carbon content changes insignificantly, but the carbon content in aromatic compounds fraction decreased, as a result of partial decomposition of mentioned aromatic compound fraction

  • Line 218: "that the changes in the metal removal efficiency were not large" - The changes not large: please specify, comparing what to what? At different PAC doses? Different contact time?

In the mentioned sentence, the changes of metal ions concentration were analysed in dependency on increased PAC dose. The information was introduced to the text.

  • Figure 2: Please write "2+" as a superscript;

Figure was corrected (actual Figure 3).

  • Figure 2: Suggest to indicate in figures themselves, which one is for 0.5 and which one for 1 mg/l. Or, mark the figures with a) and b)

Figure was corrected (actual Figure 3).

  • Line 240-241: "The extension of the process time resulted in a slight increase in the lead ..." - Isn't it so also for Ni and Cd?

We added the information that similar observation was for nickel and cadmium

  • Line 237-256: Any ideas to explain why efficiency was the highest and the most increasing namely for Pb?
    Why efficiency was better for lower initial metal concentrations?

We would like to explain that the best efficiency in lead removal was probably an effect of higher affinity of lead to used PAC when compared to other studied metals.

Better efficiency for lower metals content was observed probably because we used the same PAC doses for both metal content. Efficiency calculated as a ratio of adsorbed amount of metal (could be on similar level for both: 0.5 and 1.0 mg/L content) to its total content has to be higher for lower initial metal concentration.

  • Lines 241 and 246: "The extension of the process time resulted in a slight increase ..." / "the extension of the experiment time resulted in an increase in ..." - You are repeating what has already been said;

We removed similar sentence.

  • Line 253: "than other previously analyzed metals" - what are those other previously analyzed metals?

We changed the sentence to indicate, that another analysed metals were nickel and cadmium.

  • Table 5: Suggest to place the table after the next paragraph, and make a more clear transition from describing model water and modified water.
    Why only this PAC dose and timing are presented for modified water?

The position of table was changed and we add the sentence introducing new stage of studies.

We would like to explain, that in mentioned step of our work we decided to use the maximal values of time and PAC dose used for studies done for model water. We wanted to eliminate limitation depending on process condition (actual Table 7).

  • Lines 265 - 280: What is particular about the type of activated carbon, or about the other factors of your research? So that other researchers, or practitioners could refer to your results in their work?

We would like to explain, that at our work we use typical PAC and typical condition for water treatment. The main aim of our study was observation of combined processes usefulness for removal of typical contaminants of water prepared for human consumption and comparison to typical single processes in the context of their efficiency in organic matter and heavy metals removal

  • Line 293: "58" - It doesn't look like it is 58% in the table and the figure. You have either switched places C and C+A results there, or here in the explanation.

Thank you very much for the detailed review of our manuscript. It looks like typing  error. The mistake is already corrected in the manuscript.

  • Editing of English language is required. It is actually quite uneven throughout the text: from a fluent text to some obvious mistakes.

Additional English correction was made

Reviewer 2 Report

 I would like to mention some points about the aforementioned paper in order to be taken into account by the authors:

  1. Abstract and total manuscript

Please check English language. E.g. lines 9 and 15 some words are missing or are used in incorrect manner, (i.e. “Coagulation process the with the use of”).

The same at lines 30-31, line 100, line 237 (i.e. From among the examined heavy metals) etc.  Please check carefully the manuscript for similar errors, sometimes the meaning is missing.

  1. Introduction lines 34-35, lines 52-56, lines 78-79,

Please give relative references.

  1. Introduction

Please give the advantages and the disadvantages of each technology in a table.

  1. “polyaluminum chloride” or polyaluminium chloride?

Please correct and use throughout the manuscript the same term i.e. preferably polyaluminum chloride

  1. Results and discussion

Please also refer the respective legislation limits of selected parameters and metal ions concentration.

  1. No reference is made to the cost of each combined procedure.
  1. What is about the residual aluminum concentration that is a very important parameter that it must be taken on account when are used Al-based materials?

  1. Conclusions “The analysis of combined processes showed that in the case of pollutants such as heavy metal ions, the coagulation process had the greatest share in their removal from water.”

At this point the authors themselves respond to what is floating in my mind as I read this article: why the study all these hybrid processes to conclude that coagulation is the most effective method? Instead of using all these costly procedures in combination (to succeed in the end only up to 85% removal), the authors, could simply increase the dose of coagulant and achieve more significant results. There are also plenty of new pre-polymerized coagulants in the literature that require a lower dose and are more effective.

Author Response

We would like to thank very much for the professional, helpful an useful Reviewer comments and suggestions. We are grateful for the Reviewer’s careful reading. Correction was made according to the comments. All changes in the manuscript are indicated by yellow color.

Answer for comments of Reviewer 2:

I would like to mention some points about the aforementioned paper in order to be taken into account by the authors:

  1. Abstract and total manuscript

Please check English language. E.g. lines 9 and 15 some words are missing or are used in incorrect manner, (i.e. “Coagulation process the with the use of”).

The same at lines 30-31, line 100, line 237 (i.e. From among the examined heavy metals) etc.  Please check carefully the manuscript for similar errors, sometimes the meaning is missing.

Additional English correction was made

2. Introduction lines 34-35, lines 52-56, lines 78-79,

Please give relative references.

lines 34 – 35: reference was introduced

lines 52 – 56: references were introduced

lines 78 – 79: reference was introduced

  1. Introduction

Please give the advantages and the disadvantages of each technology in a table.

The appropriate table was introduced into the manuscript (actual Table 1)

4. “polyaluminum chloride” or polyaluminium chloride?

Please correct and use throughout the manuscript the same term i.e. preferably polyaluminum chloride

We correct inconsequence in nomenclature – we decided for British style of chemical compounds nomenclature

5. Results and discussion

Please also refer the respective legislation limits of selected parameters and metal ions concentration.

Respective legislation limits are actually presented in the manuscript (actual Table 2) in the new paragraph.

6. No reference is made to the cost of each combined procedure.

Thank you very much for valuable comment. Authors will try to take it into account during future reports preparation. We would like to explain, that in the laboratory scale the evaluation of the exact costs of process driving is very difficult (both: investment and operative costs). Due the wide range of trade active carbons, ozonators and UV lamps available, the costs of material and equipment are not as high as a few years ago.

7. What is about the residual aluminum concentration that is a very important parameter that it must be taken on account when are used Al-based materials?

 We would like to explain, that authors were studied residual aluminium concentration in water during previous studies concerning on coagulant selection for water treatment. When pre-hydrolysed salts of high-basic polyaluminium chlorides, the residual aluminium concentration was below the limit value for water for human consumption (0.2 mg/L). Also, in the presented studies residual aluminium concentration was checked during coagulation process and determined as 0.09 mg/L:

Sperczyńska E., Dąbrowska L., Wiśniowska E., Removal of turbidity, colour and organic matter from surface water by coagulation with polyaluminum chlorides and with activated carbon as coagulant aid, Desalination and Water Treatment, 2016, 57 (3), 1139-1144.

Dąbrowska L., Removal of organic matter from surface water using coagulants with various basicity, Journal of Ecological Engineering, 2016, 17(3), 66-72.

Dąbrowska L.,  The use polyaluminium chlorides with various basicity for removing organic matter  from drinking water, Desalination and Water Treatment, 2018, 134, 80-85.

8. Conclusions “The analysis of combined processes showed that in the case of pollutants such as heavy metal ions, the coagulation process had the greatest share in their removal from water.”

At this point the authors themselves respond to what is floating in my mind as I read this article: why the study all these hybrid processes to conclude that coagulation is the most effective method? Instead of using all these costly procedures in combination (to succeed in the end only up to 85% removal), the authors, could simply increase the dose of coagulant and achieve more significant results. There are also plenty of new pre-polymerized coagulants in the literature that require a lower dose and are more effective.

Thank you very much for this comment. We agree that for metals removal classical coagulation and adsorption seem to be efficient enough. But in our work we analysed organic matter removal from water also. Studied hybrid processes are mainly used for natural organic matter as well as micropollutants (e.g. pharmaceuticals, pesticides, PAHs – polycyclic hydrocarbons) degradation and removal from water. From the point of view of organic carbon content and eventual chlorination by-products formation, the additional improvement in degradation or removal of organic matter is very important.

Reviewer 3 Report

The paper is devoted to assessing the effectiveness of water treatment. A study of four hybrid processes was carried out: (1) coagulation and adsorption, (2) ozonation and coagulation, (3) ozonation, coagulation and adsorption, (4) ozonation, UV radiation, coagulation and adsorption to remove of inorganic and organic pollutants. The manuscript is well written and structured. There is a minor comment. It is recommended to cite the water quality standards adopted in the EU for studied indicators.

Author Response

We would like to thank very much for the professional, helpful an useful Reviewer comments and suggestions. We are grateful for the Reviewer’s careful reading. Correction was made according to the comments. All changes in the manuscript are indicated by yellow color.

Answer for comments of Reviewer 3:

The paper is devoted to assessing the effectiveness of water treatment. A study of four hybrid processes was carried out: (1) coagulation and adsorption, (2) ozonation and coagulation, (3) ozonation, coagulation and adsorption, (4) ozonation, UV radiation, coagulation and adsorption to remove of inorganic and organic pollutants. The manuscript is well written and structured. There is a minor comment. It is recommended to cite the water quality standards adopted in the EU for studied indicators.

Thank you very much for valuable comment. Respective legislation limits are actually presented in the manuscript (actual Table 2) in the new paragraph:

“2.4. European Union legislative limits of selected indicators of water quality”

Round 2

Reviewer 2 Report

Thank you for answering my comments